

**Occurrence and Spatial Distribution of the Neutral Per-fluoroalkyl**
**Substances, and Cyclic Volatile Methylsiloxanes in Atmosphere of the**
**Tibetan Plateau**
*Xiaoping Wang[1,2*], Jasmin Schuster[3,4], Kevin C. Jones[4], and Ping Gong[1,2]*
[1]Key Laboratory of Tibetan Environment Changes and Land Surface Processes,
Institute of Tibetan Plateau Research, Chinese Academy of Sciences, Beijing, 100101,
China
[2]CAS Center for Excellence in Tibetan Plateau Earth Sciences,Beijing, 100101, China
[3]Air Quality Processes Research Section, Environment and Climate Change Canada,
4905 Dufferin St., Toronto, ON M3H 5T4, Canada
[4]Lancaster Environment Centre, Lancaster University, Lancaster, LA1 4YQ, U.K.
**\* Corresponding author address:**
**E-mail: wangxp@itpcas.ac.cn**
**Tel: +86-10-84097120**
**Fax: +86-10-84097079**





## Abstract

Due to their properties of bioaccumulation, toxicity, and long-range atmospheric
transport, poly and per-fluoroalkylsubstances (PFASs), and cyclic volatile methyl
silxoanes (cVMS) are currently being considered as emerging persistent organic
pollutants (POPs) for regulation. To date, there are limited data on PFASs and cVMS
in the atmosphere of the Tibetan Plateau (TP), a remote environment which can provide
information on global background conditions. Sorbent-impregnated polyurethane foam
(SIP) disk passive air samplers were therefore deployed for three months (May to July
2011 and 2013) at 16 locations across the TP. Using previously reported methods for
estimating the air volume sampled by SIP disks, the derived atmospheric concentrations
ranged as follows: 18–565 ng/m$^3$ for ΣcVMS (including D3, D4, D5, and D6); 65–223
pg/m$^3$ for fluorotelomer alcohols (ΣFTOHs); 1.2–12.8 pg/m$^3$ for fluorinated
sulfonamides (ΣFOSA); and 0.29–1.02 pg/m$^3$ for fluorinated sulfonamidoethanols
(ΣFOSE). The highest ΣcVMS occurred at Lhasa, the capital city of the TP, indicating
the local contribution to the emerging pollutants. Higher levels of ΣFTOHs were
observed at sites close to the transport channel of the Yarlung Tsangpo Grand Canyon,
indicating possible long-range atmospheric transport (LRAT). Elevated concentrations
of shorter-chain volatile PFAS precursors (4:2 FTOH and fluorobutane
sulfonamidoethanol) were found in most air samples, reflecting the shift in production
from long- to short-chain PFASs in Asia. Overall, concentrations of emerging POPs at
background sites of the TP were 1–3 orders of magnitude higher than those reported for
legacy POPs.



**Introduction**


Persistent organic pollutants (POPs) have attracted significant attention due to their
wide distribution in the environment and high toxicity to humans and wildlife (Hung et
al., 2016a; Hung et al., 2016b; Magulova and Priceputu, 2016; Rigét et al., 2010). In
the first stage, the Stockholm Convention included twelve POPs, normally considered
the legacy POPs (Rigét et al., 2010), including dichlorodiphenyltrichloroethane (DDT),
hexachlorobenzene (HCB), and hexachlorocyclohexanes (HCHs). With the prohibition
of these legacy POPs, their levels in the environment have largely deceased (Hung et
al., 2016a; Hung et al., 2016b). Compared with these legacy POPs, other organic
substances, such as per-fluoroalkyl substances (PFASs), and volatile methyl-siloxanes
(VMS), have attracted more attention in recent years in the environmental chemistry
research community (Pedersen et al., 2016; Shi et al., 2015; Wang et al., 2015a; Xiao
et al., 2015) due to their widespread production, bioaccumulative behavior, and toxicity.
In 2009, perfluorooctanesulfonic acid (PFOS) and perfluorooctane sulfonyl fluoride
(POSF)-based chemicals were listed under Annex B of the restricted substances of the
Stockholm Convention (Zushi et al., 2012).
In addition, use of VMS in personal care products have also been restricted by the
European Chemical Agency. Due to their widespread use in inks, waxes, firefighting
foams, metal plating and cleaning, coating formulations, and repellents for leather,
paper, and textiles, large quantities of PFASs have been discharged into the
environment (Shoeib et al., 2006). Taking PFOS as an example, the total historical
worldwide production of "PFOS equivalent", including secondary reaction products
and precursors, was estimated to be 122,500 tons between 1970 and 2002 (Guerranti et
al., 2013; Paul et al., 2009). However, since 2002, the emission of PFASs has shifted
from North America, Europe, and Japan to emerging Asian economies, especially
China and India (Li et al., 2011; Sharma et al., 2016). Passive air sampling results have
found that fluorotelomer alcohols (FTOH) and fluorinated telomere olefins (FTO) are
major congeners occurring in the urban air of China and Japan, while 4:2 FTOH is a



predominant chemical in remote regions of China and India (Li et al., 2011).
Methylsiloxanes are widely used in industrial and commercial applications, including
additives in fuel, car polish, cleaners, waxes, and personal care products (cosmetics,
deodorants and lotions, Borga et al., 2013; Buser et al., 2013). Cyclic volatile
methylsiloxanes (cVMS) include hexamethylcyclotrisiloxane (D3),
octamethylcyclotetrasiloxane (D4), and their rearrangement products such as
decamethylcyclopentasiloxane (D5) and dodecamethylcyclohexasiloxane (D6). These
chemicals are the subject of increasing concern because of their high emissions, long
persistence (Navea et al., 2011), and toxicities (Mackay et al., 2015). D4 and D5 have
been categorized as high production volume chemicals (McLachlan et al., 2010) and
identified as new persistent and bioaccumulative chemicals in commerce (Borga et al.,
2013; McGoldrick et al., 2014). Due to the high volatility, VMS can be released into
the atmosphere during use and production (Xu et al., 2014). The half-lives of VMS in
the atmosphere range from days to weeks (Xu et al., 2014; Xu and Wania, 2013), which
allow them to undergo long-range atmospheric transport (LRAT) and arrive at remote
regions such as the Arctic and Antarctic.
Despite of the minor local emissions, remote regions can also receive pollutants by
LRAT and the contamination levels of pollutants in these areas reflect the extent to
which the remote area has been contaminated. Studies on the occurrence and
distribution of PFASs and cVMS have been conducted in various environmental media
of the Arctic (Krogseth et al., 2013) and Antarctic (Sanchís et al., 2015), where
unexpectedly high concentrations were found. As well as the Arctic and Antarctic, the
Tibetan Plateau (TP) is often referred to as the "third pole", isolated in the mid-latitude
northern hemisphere with a harsh environment and high elevation. The transport (Sheng
et al., 2013), distribution (Wang et al., 2010; 2016b), and bioaccumulation (Ren et al.,
2016) of legacy POPs in the Tibetan environment have already been investigated;
however, there is still a gap in knowledge regarding the distribution of emerging
organic contaminants, such as PFASs and cVMS.





In this study, sorbent-impregnated polyurethane foam (SIP) disk passive air samplers
were deployed across the TP (16 sites) to obtain the spatial distribution of PFASs and
cVMS in the atmosphere. These sites include densely populated cities and background
sites, in order to test how the local emission and LRAT contaminate the TP. Combining
the results of this study with the published data regarding legacy POPs in the TP, and
emerging POPs in other Asian regions, will provide useful insights to help rank the
exposure risks of legacy and emerging POPs in the Tibetan environment, and gain a
comprehensive understanding of the distribution pattern of emerging POPs in Asia.
**Materials and methods**
**Preparation of SIPs.** Air monitoring in remote areas is especially challenging due to
the lack of electricity. Passive air samplers (PAS) have the advantage that they do not
require electricity, and are also cheap and easy to handle. Among the various PAS, SIP
uses polyurethane foam (PUF) coated with polystyrene divinylbenzene copolymeric
resin (XAD-4) as the absorption medium, which has been widely used for a range of
POPs, including PFASs, VMS, and PCBs (Ahrens et al., 2013; Genualdi et al., 2010,
2011; Shoeib et al., 2008). The preparation of SIP was conducted at Lancaster
University, U.K., following the previously published method (Shoeib et al., 2008).
Briefly, PUF disks (Tisch Environmental) were pre-extracted in a Soxhlet with acetone
(12 h) and petroleum ether (18 h). Amberlite XAD-4 was precleaned by sonication in
methanol, dichloromethane, and hexane (30 min each). The precleaned Amberlite
XAD-4 was ground to a powder using a Retsch planetary ball mill (particle diameter
approximately 0.75 μm). The PUF disks were coated with the XAD-4 by dipping the
precleaned disks in a dispersion of the powdered Amberlite XAD-4 slurry in hexane.
SIP-PUF disks were dried under vacuum, and an average of $435 \pm 30$ mg of XAD-4
coated each disk ($n = 80$; each sampling had 32 samples and 8 field blanks), which was
similar to the Global Passive Atmospheric Sampling program (Genualdi et al., 2010).
All prepared SIP disks were stored in sealed metal tins at $-17°C$ until they were
transferred to the sampling locations.



**Sampling campaign**


Taking advantage of the Tibetan Observation and Research Platform (Wang et al.,
2016a), a passive air monitoring network comprising 16 sampling sites across the TP
has been established, with good spatial coverage of the TP (Supporting Information,
Fig. S1), and has already produced results regarding the spatial and temporal pattern of
legacy POPs (Wang et al., 2010; 2016b). In this study, duplicate SIP-PAS were
deployed at each sampling site for about 100 days from May to July for sampling
PFASs (2011) and cVMS (2013). During the sampling, another PUF sampler was co-
deployed to obtain the site-specific sampling rate using four depuration compounds
(DCs; PCB-30, -54,-104, and -188; Pozo et al., 2009). Details relating to the DCs can
be found in Text S1. The sampling program and meteorological conditions at each site
are provided in Table 1. Field blanks were unpacked and exposed them in air for 1 min
at the sampling site and then treated as real samples. At the end of the deployment
period, the collected SIP-PUF and PUF disks were sealed in metal tins and transported
to the clean lab in Lhasa for extraction.

**Sample extraction and analysis**


The target PFASs were neutral PFASs, including fluorotelomer olefin (8:2 FTO),
fluorotelomer acrylates (6:2, 8:2 FTA), fluorotelomer alcohols (4:2, 6:2, 8:2, 10:2, and
12:2 FTOH), sulfonamides (NMeFBSA, NMeFOSA, and NEtFOSA), and
sulfonamidoethanols (NMeFBSE, NMeFOSE, and NEtFOSE); the four target cVMS
were D3, D4, D5, and D6. PFAS standards were purchased from Wellington
Laboratories Inc. (Guelph, Ontario, Canada), and D3, D4, D5, and D6 were purchased
from Tokyo Chemical Industries America (Portland, OR).
Extraction of the PFASs was performed by sequential cold column extraction with ethyl
acetate as the extraction solvent. Field blanks and lab blanks were extracted along with
samples in the same way. After the spiking of the recovery standard (see Table S1 for
the composition), SIP was extracted by three separate immersions (30 min) in ethyl



acetate, and all three extracts were combined and concentrated. These extracts were
then filtered by Millipore Millex syringe filter unite (0.45μm, 4mm), reduced to a
volume of 1 ml and cleaned up by 2 cm of Envi-Carb. Finally, after adding the internal
standard (Table S1), the extracts were reduced to 50 μl for injection. The analysis of
volatile PFASs was performed using GC-MS equipped with a SUPELCO WAX
column (60 m, 0.25 mm inner diameter, 0.25 μm film, Supelco, Bellefonte, PA) under
positive chemical ionization mode.
Before sampling, the SIP disks were spiked with recovery mixture containing each of
the $^{13}C_4$-D4, $^{13}C_5$-D5, and $^{13}C_6$-D6, and after sampling, they were Soxhlet extracted
with petroleum ether/acetone (85/15, v/v) for around 6 h. All extracts were then
concentrated by rotary evaporation, followed by gentle nitrogen blow-down to 0.5 mL
using isooctane as a keeper for the extracts. Mirex was added to the final extract as an
internal standard. The separation and detection of the cVMSs was performed using GC-
MS in selective ion monitoring mode using a DB-5 column (60 m, 0.25 mm inner
diameter, 0.25 μm film, J&W Scientific).
**Quality assurance/quality control**
Samples were extracted in a clean lab with filtered, charcoal stripped air and positive
pressure conditions. All glassware used for sample collection was cleaned, and baked
at 450°C before use. Powder-free nitrile gloves were used for all handling of the
samples. All personnel involved in sample collection and analysis refrained from using
personal care products to avoid contamination. A total of eight field blanks and six lab
blanks were analyzed for target PFASs. In the lab blanks, only 8:2 FTOH and 10:2
FTOH were screened, which showed low concentrations, while 4:2 FTOH, 8:2 FTOH,
10:2FTOH, NEtFOSA, NMeFOSE, and NEtFOSE were observed in field blanks, with
concentrations ranging between 50 and 321 pg/sample (Table S2). Similarly, eight field
blanks and six lab blanks were arranged for evaluating the uncertainties of cVMS
concentrations due to contamination and loss processes (during the extraction and
cleanup procedures and storage). D3, D4, D5, and D6 in field blanks were, on average,



34, 57, 380, and 59 ng/sample, respectively, which were approximately 6% of the
sample concentration. Method detection limits (MDLs) were calculated from the blanks
[average of blanks + 3 × standard deviation ( σ )]. Based on this principle, MDLs of
volatile PFASs ranged between 37 and 419 pg/sample, while MDLs of cVMS ranged
between 52 and 681 ng/sample (Table S3). Details of the MDLs for each congener are
provided in Table S2 and S3.
The average recoveries were $88 \pm 27\%$, $79 \pm 34\%$, $71 \pm 27\%$, $95 \pm 21\%$, and $107 \pm 19\%$
for 5:1 FTOH, 7:1FTOH, [M+5]8:2 FTOH, 9:1 FTOH, and 11:1 FTOH, respectively;
and $117 \pm 33\%$, $105 \pm 27\%$, $89 \pm 37\%$, $93 \pm 33\%$, and $92 \pm 29\%$ for [M+3]NMeFOSA,
[M+5]NEtFOSA, [M+7]NMeFOSE, and [M+9]NEtFOSE, respectively. These
recoveries were broadly in line with previous passive air sampling for Asian counties
in which the same SIP disks were deployed (Li et al., 2011). The recoveries were 116.0
$\pm$ 5.9%, $90 \pm 8.5\%$, and $98.2 \pm 1.7\%$ for $^{13}$C-D4, $^{13}$C-D5, and $^{13}$C-D6, respectively.
Recoveries over 100% were observed for $^{13}$C-D4, which may be due to transformation
to $^{13}$C-D4 from $^{13}$C-D5 during sampling (~100 days) and storage.

**Sampling rate calculation**

Generally, the uptake profile of a chemical to the passive sampler medium (PSM)
includes three stages: 1) quick, linear uptake when the amount of chemicals in the PSM
is small; 2) curvilinear uptake (slow uptake); and 3) equilibrium uptake when the
amount of chemicals in the PSM reaches a plateau. Volatile compounds usually have
short linear phase absorption and equilibrate after a few weeks in SIP (Ahrens et al.,
2013; Shoeib et al., 2008), while longer linear phases will occur if SIP is operated at
colder temperatures (Ahrens et al., 2013). In a previous calibration study (in which the
sampling temperature was 18°C), linear phase uptake of PFASs in SIP was reported
(Ahrens et al., 2013), due to the greater capacity of SIP-PAS to PFASs. However, the
sampling temperature in the present study (Table 1) was much lower, and so linear
phase absorption should be expected to occur. For this reason, the previously reported
average linear sampling rate ($R$) of 4 m$^3$d$^{-1}$ reported by Ahrens et al (2013) for PFASs



(including FTOHs, FOSAs and FOSE) was chosen to estimate the final sample air
volume of the SIP-PAS (multiplying 4 $m^3 d^{-1}$ by the number of days of deployment).
Based on this estimation, volumetric concentrations of target compounds were obtained
and are presented in Table S4. The MDLs in Table S4 were also calculated based on
the 90-day exposure duration.
The volume of air sampled for cVMS in SIP disks can be described by the following
equation:
$V_{air} = K_{SIP-A} \times V_{SIP} \times (1 - \exp\{-(A_{SIP})/(V_{SIP}) \times (k_A/K_{SIP-A})\}t)$        (1)
where $V_{air}$ is the air volume sampled by the SIP disk, $K_{SIP-A}$ is the SIP-air partition
coefficient, $V_{SIP}$ is the volume of the SIP disk ($cm^3$), $A_{SIP}$ is the planar surface area of
the SIP disk ($cm^2$), $k_A$ is the air-side mass transfer coefficient (m/day), and $t$ is
deployment time (days). $K_{SIP-A}$ is highly temperature dependent and can be calculated
using its correlations with $K_{OA}$ (Ahrens et al., 2014). Details about the calculation are
presented in Table S5. Values of $k_A$ can be derived from the site-specific sampling rate
($Rs$) and the surface area of the SIP disk ($A_{SIP}$). The $Rs$ values were calculated from the
use of DCs on the PUF disks that were co-deployed at each site. Details of these
calculations have been previously reported and are presented in Text S2 and Table S6.
The values of log ($K_{SIP-A}$) for D3, D4, D5, and D6 are listed in Table S7; and the air
volume sampled by the SIP disk are provided in Table S8. Then, volumetric
concentrations of D4, D5, and D6 are presented in Table S9.
**Result and discussion**
**Concentration of neutral PFASs and cVMS**
From Table S5, with the exception of fluorotelomer acrylates (6:2, 8:2 FTA), all neutral
PFAS congeners were quantitatively detected in all samples. This implies that the
neutral PFAS were ubiquitous in the air of the TP. The dominant compounds were FT
alcohols, with the total concentration of FTOH (sum of 4:2 FTOH, 6:2 FTOH, 8:2





FTOH, 10:2 FTOH and 12:2 FTOH) ranging from 65 to 223 pg/m$^3$. These values are
lower than those measured in Chinese cities, such as Beijing, Taiyuan, and Changsa (Li
et al., 2011) but are higher than those reported at background sites, including remote
mountains in China (80–120 pg/m$^3$, Li et al., 2011), Antarctica (13.5–46.9 pg/m$^3$, Wang
et al., 2015b), and the Arctic (7.7–49 pg/m$^3$, Shoeib et al., 2006). Among all the FTOHs,
concentrations of 8:2 and 4:2 FTOH were the highest, being in the tens of (up to one
hundred) pg/m$^3$. Generally, 8:2 FTOH was the prevailing compound found in the gas
phase. This may be due to its relatively high volatility and long atmospheric lifetime
(Rayne et al., 2009). However, concentrations of 8:2 FTO were in the range of 0.88 to
4.56 pg/m$^3$, which are lower than those measured in other background regions (~ tens
of pg/m$^3$, Li et al., 2011). Levels of fluorinated sulfonamides (sum of NMeFBSA,
NMeFOSA, and NEtFOSA) in Table S4 can reach a maximum of around 10 pg/m$^3$,
while the total concentration of sulfonamidoethanols (including NMeFBSE,
NMeFOSE, and NEtFOSE) was only a few pg/m$^3$, which is an order of magnitude
lower than those observed for sulfonamides. It is clear that the proportion of FTOHs
was much higher than FOSEs and FOSAs, which may be due to FOSEs and FOSAs
being prone to absorption on particles (Li et al., 2011).
The measurements reported here represent the first survey of concentrations of cVMS
in the TP (also known as "the Third Pole", Qiu, 2008). All measured cVMS
concentrations were above the MDL, suggesting cVMS were also ubiquitous in the
Tibetan atmosphere (Table S6). The average atmospheric concentrations for D3, D4,
D5, and D6 were 29.1, 38.8, 88.6 and 1.6 ng/m$^3$, respectively (Table S9).
Concentrations of D5 were higher than D3 and D4, which is different from the reported
concentrations of 17, 16, 4.0, and 0.54 ng/m$^3$ for D3, D4, D5, and D6 at the Zeppelin
observatory (Arctic) using the same SIP-disks for sampling (Genualdi et al., 2011).
However, similar to other Arctic results, D5 was the dominant congener in air (Krogseth
et al., 2013). D5 is the most frequently used cVMS in personal care products, and
therefore is the predominant cVMS in the urban atmosphere (McLachlan et al., 2010).
However, dominance of D5 have been observed in both Arctic and Antarctic region,



highlighting its persistence in atmosphere and LRAT potential. The obtained cVMS concentrations in the TP are higher than those reported for Arctic and remote Sweden, indicating the possible local contamination. Both PFASs and cVMS are closely associated with human activities, and their concentrations usually show positive correlations with population (Genualdi et al., 2010; Nguyen et al., 2016). Therefore, we would expect high concentrations of volatile PFASs and cVMs in the atmosphere of Lhasa and Golmud, which are the two largest cities on the TP, with relatively large populations and fast urbanization. From Table S4 and S9, in Lhasa (the capital and also the largest city of the Tibet autonomous region), the concentrations of 8:2 FTOH and 4:2 FTOH were 71 and 43 pg/m$^3$, respectively, and similar levels were also found for Golmud. Additionally, concentrations of D5 in Lhasa and Golmud were 465 and 208 ng/m$^3$ respectively, which were the two highest D5 concentrations in the Tibetan atmosphere (Table S10). Although these levels were still orders of magnitude lower than those reported for other megacities (Genualdi et al., 2010; Mackay, 2015) the elevated concentrations suggest that the expansion/development of cities, followed by the migration of rural populations, may lead to an increase of emerging pollutants in Tibet.

**Spatial distribution and congener profile of neutral PFASs**

An important objective of this study was to improve knowledge on the spatial patterns of emerging POPs in the background air across the TP. In previous studies, the spatial distributions of atmospheric legacy organochlorine pesticides over the TP have been reported, and were found to be strongly related to the air circulation patterns of the TP, i.e. the Indian Monsoon and westerly winds (Figure S2, Wang et al., 2010; 2016b). For example, DDT-related chemicals were major chemicals in the atmosphere of the southeastern TP, which is influenced by the Indian monsoon air masses (Wang et al., 2010); whereas, the northwestern TP was dominated by HCB in the atmosphere, caused by the westerly-driven European air masses (Wang et al., 2016b). Similarly, ice cores collected in different regions of the TP indicated that PFOS existed in the Muztagata glacier (western TP); while in the Zuoqiupu glacier, located in the eastern TP, PFOS



was below the detection limit, but concentrations of short-chain perfluorobutanoic acid
have increased during recent years (Wang et al., 2014). All these results suggest that
differences in the concentrations and composition profiles of POPs likely reflect the
upwind sources affecting the different parts of the TP (e.g., European/central Asian
sources for the west TP and Indian sources for the east TP).
Figure 1 presents the spatial patterns of 8:2 FTO, FTOHs, FOSAs, and FOSEs. The
spatial distribution of 8:2 FTO shows a decreasing gradient from the east to the west of
the TP (Figure 1). On the basis of the ANOVA results, significantly high values of 8:2
FTO were found at Qamdo and Bomi (Table S10). However, spatial variation was
found in total FTOHs (Figure 1), and significant differences only occurred at the east
regions (Chayu, Rawu, and Lulang) and the western sites (Gar and Muztagata, Table
S7). It is noted that the highest $\Sigma$FTOH concentration occurred at Chayu (222 pg/m$^3$),
which is on the southern slopes of the Himalaya and close to the China–India border.
Levels of $\Sigma$FTOHs in Chayu were even higher than that of Lhasa (180 pg/m$^3$),
suggesting that the southeast part of the TP may receive considerable inputs of PFASs
from south Asia. Regarding $\Sigma$FOSAs and $\Sigma$FOSEs, higher levels were seen in both the
east and west of the TP (Figure 1), compared to the middle of the TP. A previous study
observed high levels of atmospheric DDTs at sites (e.g. Chayu, Rawu, Bomi, etc.) close
to the Yarlung Tsangpo Grand Canyon (Wang et al., 2016b). Here, $\Sigma$FTOHs, $\Sigma$FOSAs,
and $\Sigma$FOSEs also showed higher levels at these sites (Figure 1), which confirms
previous results that show that the Yarlung Tsangpo Grand Canyon is a channel for
receiving pollutants from southern Asia (Sheng et al., 2013; Wang et al., 2016b).
Medium $\Sigma$FOSA and $\Sigma$FOSE concentrations found in the Muztagata region broadly
agree with the previous results that air masses originating from European sources are
generally clean (Wang et al., 2016b).
As mentioned above, the composition profile of POPs is closely associated with air
circulation patterns in the TP and can reflect the upwind sources. However, congener
profiles of neutral PFASs in this study did not show any clear difference between
western sites (e.g. Muztagata, Gar) and eastern sites (Chayu, Bomi, Lulang, etc.)



(Figure 1), which may be because the sampling period was too short (~ 3 months) and
only covered the monsoon season (June to September). Elevated 4:2 FTOH and
NMeFBSE concentrations were found in most of the samples of the present study and
a dominance of shorter-chain volatile PFAS precursors was the feature of the south
Asian sources (Li et al., 2011). This similarity suggests that neutral PFASs in the TP
may originate mainly from south Asia, most likely by LRAT.
Although the congener profiles cannot be used to distinguish the European and Indian
sources in this study, the ratio of 8:2 to 10:2 to 6:2 FTOH is an excellent indicator of
LRAT for atmospheric PFASs (Wang et al., 2015b). The transport fate of atmospheric
PFASs can be influenced by photochemical degradation. A higher ratio indicates the
aged nature of the air mass because of the fast photochemical degradation of 6:2 FTOH
(half-life = 50 days) in the air compared with 10:2 FTOH (70 days) and 8:2 FTOH (80
days, Piekarz et al., 2007). During LRAT, more 6:2 and 10:2 FTOH will be removed
from the atmosphere. For example, ratios of 6.4:2.1:1.0 were observed in the Arctic
(Ahrens et al., 2011) and 35.6:6.5:1.0 were found in the Antarctic (Wang et al., 2015b).
In the present study, low ratios were observed in the cities, i.e. 2.4:1.7:1 and 6.8:1.2:1
were observed for Lhasa and Golmud, respectively. This indicates that cities are
possible fresh emission sources of neutral PFASs. According to a previous study, there
are three climate zones over the TP—namely, the monsoon region, westerly region, and
transition region (Wang et al., 2016b). The sampling sites of this study can be grouped
into these three zones (Figure S2, Table S11). The average ratios of 8:2 to 10:2 to 6:2
FTOH were 8.4:1.2:1 for the monsoon region, 8.8:1:1 for the westerly region and
10.6:1.2:1 for the transition zone (Figure S2, Table S11). Overall, these values are
comparable to those reported for the Arctic. On the other hand, a decrease in 6:2 FTOH
and an increase in 8:2 and 10:2 FTOH occurred from the edge regions to the central
part of the TP (Table S11). The high ratios indicate the aged nature of atmospheric
PFASs in the atmosphere of the TP, especially around the transition zone (Table S11).
Given that the transition zone is located in the hinterland (central part) of Tibet,
relatively far away from the source regions of either India or Europe/central Asia, the



aged PFASs in the air of central TP is expected and reasonable.
**Correlations between PFAS compounds**
Correlations between concentrations of pollutants can be used to test if they have some
common sources or undergo similar environmental fates. A correlation matrix was
therefore prepared and showed that some chemicals were significantly correlated
(Table 2). Good correlations ($r > 0.80$, $p < 0.01$) were observed between 8:2 FTOH and
10:2 FTOH ($r = 0.90$), and between 10:2 FTOH and 12:2 FTOH ($r = 0.97$). This
phenomenon has been observed in other studies (Ahrens et al., 2012; Cai et al., 2012;
Li et al., 2011) and usually suggests that 8:2 FTOH, 10:2 FTOH, and 12:2 FTOH have
the same source. Correlations between 4:2 FTOH and other FTOHs are generally low,
or not significant, indicating 4:2 FTOH may come from different sources. There is
much evidence that the manufacture of PFASs has shifted from longer-chain chemicals
(C8 or above) to shorter-chain ones (Butt et al., 2010; Hogue, 2012), which may lead
to the poor correlation between 4:2 FTOH and other FTOHs. Given the new production
of shorter-chain PFASs mainly centered in Asian countries, such as China and India
(Hogue, 2012) it is not surprising that high levels of both 4:2 FTOH and its independent
characteristics have been found in the Tibetan atmosphere, due to the close proximity
between Tibet and south Asia.
With regard to the relationships between FOSAs and FOSEs, good correlations were
seen among NMeFBSA, NMeFOSA, and NEtFOSA (Table 2). Additionally,
concentrations of NMeFBSE were significantly correlated with those of NMeFOSE
(Table 2). This is in contrast to previous results, in which poor relationships were found
between short- and long-chain PFASs (Li et al., 2011). Regarding the emission patterns
of FOSAs and FOSEs in India, mixed manufacturing with both extensive emissions of
NMeFOSA and NMeFOSE, coupled with wide discharge of NMeFBSA, have been
reported in the Indian environment (Li et al., 2011). This indicates that both long- and
short-chain PFAS are produced in south Asia. Favored by the transport of the Indian
monsoon, the co-transport of short- and long-chain PFASs may lead to a blending of



these chemicals in the Tibetan air. Meanwhile, the two short-chain PFASs, 4:2 FTOH
and NMeFBSA, were significantly ($r = 0.84$, $p < 0.01$; Table 2) correlated with each
other, suggesting these precursors may be released together in the source region.

**Spatial distribution of cVMS across the TP**


As mentioned above, greater levels of cVMS were found in the urban areas of Lhasa
and Golmud. This can also be seen in the spatial map of cVMS (Figure 2). However,
high levels of cVMS also occurred in the remote southeast of Tibet (Figure 2). Unlike
the spatial pattern of neutral PFASs, concentrations of cVMS decreased from southeast
to northwest TP (Figure 2, Table S9). Although there are no studies that report the
cVMS levels and patterns in south Asian countries, due to the source of cVMS to the
environment taking place via the use of personal care products we can expect the
regions of south Asia (e.g. the Indo-Gangetic Plain), with its high population density,
to be important cVMS source regions. The close proximity of the southeast TP to south
Asia and the fast LRAT potential of cVMS (Xu et al., 2014; Xu and Wania, 2013) might
be the reason that high concentrations of cVMS occur in the southeast TP. On the other
hand, latitude might be a factor representing the influence of the emission source on the
spatial pattern.
Another reason that can also influence the atmospheric concentration of cVMS is their
atmospheric degradation by hydroxyl radicals. In the Arctic, low levels of hydroxyl
radicals during the polar night promotes the accumulation of cVMS in the air, while the
polar day enhances the degradation, causing the strong seasonality of cVMS in the
Arctic (Krogseth et al., 2013). The polar day usually increases hydroxyl radicals in the
air and enhances the photo-degradation of contaminants (Krogseth et al., 2013). The
level of hydroxyl radicals is generally proportional to the extent of solar UV radiation
(Rohrer and Berresheim, 2006). Recently, Liu et al. (2017) published two UV radiation
datasets that cover the whole of China, and high values were observed for the south TP,
with a gradual decrease from the south to the north TP. Although the sampling sites in
their study were not exactly the same as in our study, their spatial trend of UV radiation





suggested that latitude might be a possible proxy to describe the variation of UV
radiation over the TP. Additionally, from a global perspective, surface UV radiation
increases with elevation due to the shorter distance of travel through the atmosphere
(Sola et al., 2008), which may also have a negative influence on the atmospheric
concentration of cVMS. Thus, elevation and latitude can be integrated together to
simulate the effects of UV radiation (representing the influence of hydroxyl radicals)
on concentrations of cVMS. On the other hand, latitude is also a factor that can
represent the influence of emission sources; low-latitude regions will receive more
cVMS due to their proximity to source regions (see Figure 2). Thus, an empirical model
was derived here to estimate the combined effects of UV radiation and the distance to
emission source regions on concentrations of cVMS:
$$C_{cVMS} = a + b\ Elevation + c\ Latitude \quad (2)$$
where $a$, $b$ and $c$ are coefficients determined from statistical regression. For the multiple
linear regressions, the $R^2$ values can be used to explain the variation of the dependents.
According to the correlations (the data from Lhasa and Golmud were excluded), the
relationship can be described as in the following:
$$C_{cVMS} = 134 - 0.011\ Elevation - 2.35\ Latitude \quad (R^2 = 0.60, p < 0.01)$$
This means elevation and latitude can jointly explain 60% of the atmospheric
concentration of cVMS. Other factors, such as cloud coverage and sky clarity (which
influence hydroxyl radical levels in the air), may be the confounding factors that
influence the correlation (Sola et al., 2008). The slope for elevation ($b$) is negative,
suggesting that high concentrations of cVMS will occur at sites with low elevation,
where hydroxyl radiation is limited. Two competing factors influence the coefficient
for latitude. The contribution from the proximity to source regions means that the low-
latitude regions of the TP will have high concentrations of cVMS (negative correlations
between latitude and $C_{cVMS}$), due to these sites being close to the source regions of south
Asia, while the strong hydroxyl degradation caused by UV radiation at low latitudes





would have the opposite effect of reducing the concentrations of cVMS (positive
correlation between latitude and $C_{cVMS}$). From the above model, the slope for latitude
($c$) in the model is also negative ($-2.35$), implying that the contribution from the
proximity to source regions to concentrations of cVMS is broadly greater than that of
hydroxyl degradation.

**Correlations between cVMS congeners**

Similar to previously published studies, good correlations were found between D3, D4,
and D5 (Table S9). The correlation coefficients varied from 0.69 to 0.79 (all
correlations were significant at the 95% confidence level; the data from Lhasa and
Golmud were excluded), while the correlation between D5 and D6 was not significant.
The good correlation implies that either D3, D4, and D5 have common sources and
transport mechanisms, or there is chemical transformation to D3 and D4 from D5
(Kierkegaard et al., 2010).

**Comparison of Measured and Modeled D5 Concentrations.**

The measured D5 concentrations are compared with the concentrations predicted by the
Danish Eulerian Hemispheric Model (DEHM, McLachlan et al., 2010). The country-
based emissions were distributed into the DEHM grid according to a data set of the
gridded population density of the world with the total emission of D5 within the DEHM
model domain estimated as 30 kT per year (McLachlan et al., 2010). All physical-
chemical properties of D5 used in model prediction are reported in previous study
(Brooke et al., 2009; Jiménez et al., 2005). The rate constant for the reaction of D5 with
OH radicals measured by Atkinson (1991) was employed. NCEP (National Centers for
Environmental Prediction) global analysis meteorological data are used to driven model.
By comparing different scenarios, the DEHM model found that phototransformation is
the dominant elimination process between emission of the D5 and arrival at the
sampling site. There is good agreement between the spatial variability in D5
concentration between the measurements of the TP and the model prediction,



displaying great D5 concentrations in southeast TP. The good tracking of the measured
concentration by the DEHM shows that D5 is clearly subject to LRAT, although it is
also effectively removed from the atmosphere via phototransformation. However,
measured D5 concentrations are 1-3 magnitudes higher than the model prediction.
Given atmospheric emission data of D5 in DEHM are estimated from usage of
antiperspirant and skin creams, the emission uncertainties might lead to the discrepancy
between measured concentrations and model values.
**Implications**
To the best of our knowledge, this is the first study on atmospheric concentrations of
neutral PFASs and cVMS in the TP region. Due to the remoteness of the TP, the
contamination of these emerging compounds will provide insight into how and to what
extent the emissions in the source regions influence these last pieces of pristine land.
Levels of neutral PFASs in the air of the TP are in the hundreds of $pg/m^3$, and levels of
cVMS are in the $ng/m^3$ range. These values are 2–3 times and 1–2 orders of magnitude,
respectively, higher than those for legacy chemicals (such as DDT and HCHs, with
maximum concentrations in the tens of $pg/m^3$, Wang et al., 2016b). Moreover, among
the various legacy and emerging POPs in wild Tibetan fishes, the average level of
ΣPFASs is the third highest (just after those of ΣDDT and ΣHCHs, Shi et al., 2015;
Wang et al., 2016a). All this evidence suggests that emerging POPs should be of great
concern for the environmental safety of the TP, as they are large volume production
chemicals that have not been regulated in the surrounding countries. Due to the LRAT
potential of volatile PFASs and cVMS, joint regulation of these emerging chemicals by
south Asian counties (upwind of the TP) has been requested in order to protect the
Tibetan environment. Taking data from this study and the pilot study for Asian
countries (Li et al., 2011) into account, due to the growing population and the transfer
of production factories from developed countries to Asian counties, Asian cities will
increasingly be the sources of emerging POPs from a global perspective.
China has not strongly regulated the manufacture of PFASs or the use of personal care



products. Over the last ten years, extensive urbanization has occurred in China. For
example, the population in Lhasa reached 90,000 in 2015, having increased by 33%
from 2014. It is estimated that the population in Lhasa will reach 110,000 in 2020. Thus,
emissions of emerging compounds due to urbanization will inevitably increase.
Following the population expansion, wastewater treatment plants deployed in cities will
not only emit volatile PFASs and cVMS into the air, but will also contaminate the TP's
water bodies (i.e. rivers, wetlands, and lakes), which are precious clean water resources.
Thus, the risks posed by city expansion to the burden and transport of pollutants should
be of great concern. Increasingly, concern regarding the toxicity and exposure risks of
PFASs and cVMS is growing among scientists and regulators. This work has important
implications for policymakers in comprehensively protecting the Tibetan alpine
environment and promoting sustainable development in Tibet (the water tower of Asia).
**Acknowledgements.** This study was supported by the National Natural Science
Foundation of China (41671480 and 41222010), Youth Innovation Promotion
Association (CAS2011067) and the International Partnership Program of the Chinese
Academy of Sciences (Grant No. 131C11KYSB20160061).

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



**Table 1 Description of the sampling program**

| Sampling site | Longitude | Latitude | Elevation /m; Temperature/°C | Description | Date of sample collection 2011 | Date of sample collection 2013 |
|---|---|---|---|---|---|---|
| Bomi | E 95°46.167' | N 29°51.485' | 2720; 8.8 | Hydrological observation station, remote area | 05/02–07/28 | 05/05–07/25 |
| Rawu | E 96°54.745' | N 29°22.289' | 4540; -2 | Rural site, 20 km from Rawu Lake | 05/03–07/31 | 05/05–07/26 |
| Lunang | E 94°44.246' | N 29°45.908' | 3330; 5.4 | Meteorological station in forest region, remote area | 05/02–07/28 | 05/05–07/31 |
| Qamdo | E 97°08.624' | N 31°09.014' | 3250; 7.6 | Rural site, 50 km from farm land | 05/04–07/31 | 05/06–07/28 |
| Chayu | E97°29.4' | N 28°37.2' | 1400; 12.4 | Meteorological station, remote area | 05/05–07/31 | 05/02–07/29 |
| Nam Co | E 90°57.800' | N 30°46.375' | 4740; -2.2 | Meteorological station near the Nam Co lake, remote area | 05/05–07/25 | 05/05–07/31 |
| GBJD | E 93°14.478' | N 29°53.122' | 3420; 6.2 | Hydrological observation station, remote area | 05/03–07/28 | 05/04–07/28 |
| Lhasa | E 91°01.956' | N 29°38.728' | 3660; 8.1 | Building roof of the Lhasa campus | 05/01–07/31 | 05/08–07/28 |
| Lhaze | E 87°38.094' | N 29°05.405' | 4020; 6.8 | Meteorological station, rural site | 05/02–07/31 | 05/04–07/27 |
| Xigaze | E 88°53.319' | N 29°15.014' | 3840; 6.6 | Meteorological station, rural site | 05/03–07/31 | 05/05–07/24 |
| Mt. Everest | E 86°56.948' | N 28°21.633' | 4300; 4.3 | Meteorological station near the Mt. Everest, remote area | 05/02–07/31 | 05/03–07/29 |
| Saga | E 85°13.951' | N 29°19.889' | 4500; 6.5 | Rural site and without agriculture activities | 05/07–07/25 | 05/06–07/28 |
| Golmud | E 94°54.480' | N 36°23.637' | 2830; 5.3 | Observation station for frost soil, rural site | 05/02–07/27 | 05/06–07/27 |
| Naqu | E 91°58.827' | N 31°25.373' | 4500; -1 | Hydrological observation station, remote area | 05/02–07/31 | 05/05–07/26 |
| Gar | E 80°05.654' | N 32°30.116' | 4300; 0.6 | Meteorological station, remote area | 05/06–07/31 | 05/03–07/27 |
| Muztagata | E 74°50.919' | N 38°16.072' | 5200; -6 | Meteorological station, remote area | 05/09–07/31 | 05/07–07/29 |







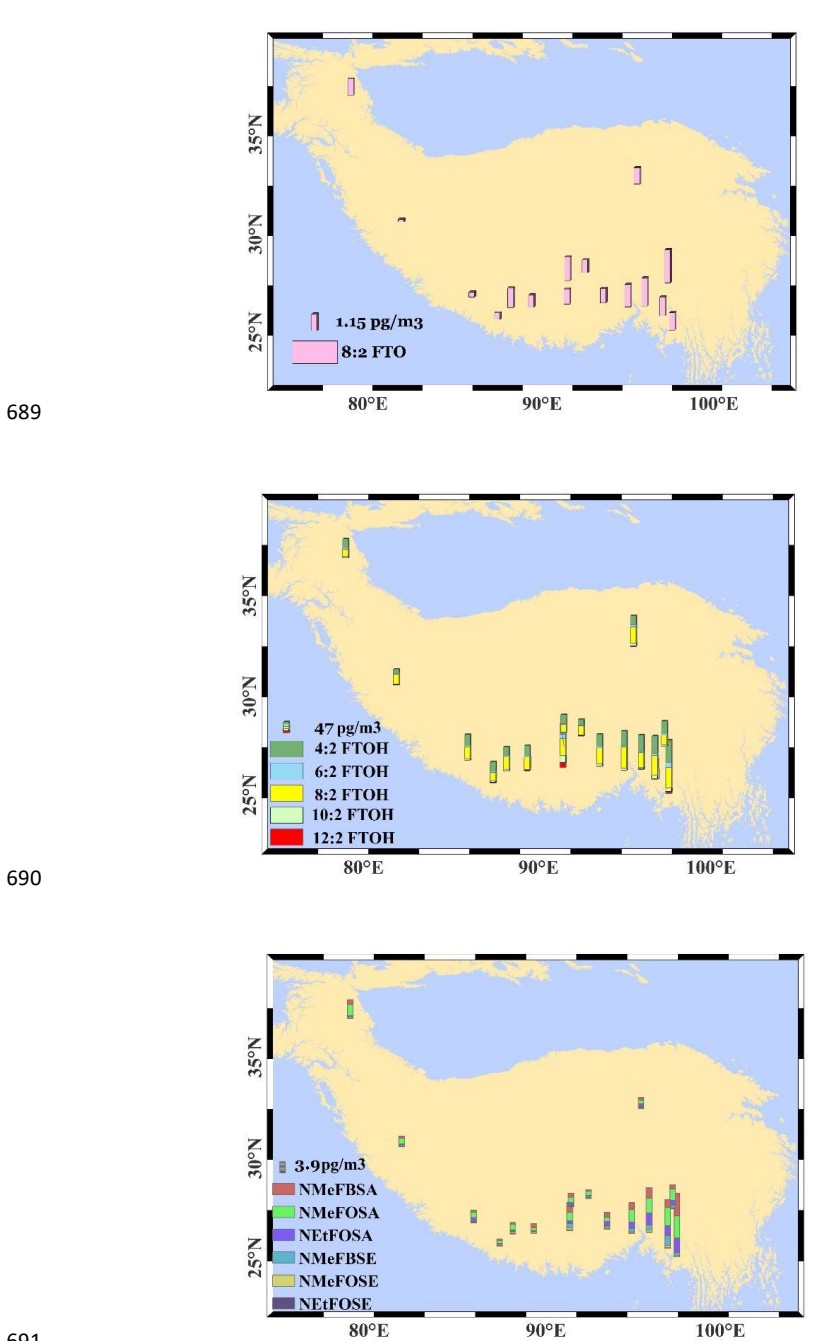




**Figure 1 Spatial distribution of neutral PFASs in the atmosphere of the TP**



**Table 2 Correlation (*r*) of individual compounds among all the samples**

| | 4:2 FTOH | 6:2 FTOH | 8:2 FTOH | 10:2 FTOH | 12:2 FTOH | NMeFBSA | NMeFOSA | NEtFOSA | NMeFBSE | NMeFOSE | NEtFOSE |
|---|---|---|---|---|---|---|---|---|---|---|---|
| 8:2 FTO | 0.44 | 0.12 | 0.18 | 0.04 | 0.00 | 0.32 | 0.46 | 0.44 | 0.37 | 0.20 | −0.11 |
| 4:2 FTOH | | **0.62** | 0.49 | 0.37 | 0.25 | **0.84** | **0.84** | **0.92** | *0.56* | 0.42 | −0.17 |
| 6:2 FTOH | | | **0.68** | *0.59* | **0.84** | **0.84** | *0.60* | *0.57* | 0.39 | *0.62* | −0.32 |
| 8:2 FTOH | | | | **0.90** | 0.45 | **0.63** | *0.57* | *0.67* | *0.58* | **0.63** | −0.33 |
| 10:2 FTOH | | | | | **0.97** | *0.58* | 0.35 | 0.30 | 0.42 | **0.77** | −0.21 |
| 12:2 FTOH | | | | | | *0.52* | 0.26 | 0.19 | 0.33 | **0.71** | −0.14 |
| NMeFBSA | | | | | | | **0.83** | **0.84** | 0.44 | 0.42 | −0.24 |
| NMeFOSA | | | | | | | | **0.88** | **0.69** | 0.43 | −0.24 |
| NEtFOSA | | | | | | | | | **0.63** | 0.37 | −0.03 |
| NMeFBSE | | | | | | | | | | **0.75** | −0.03 |
| NMeFOSE | | | | | | | | | | | −0.13 |

Bold and italic are significant at $p < 0.01$ and $p < 0.05$, respectively.



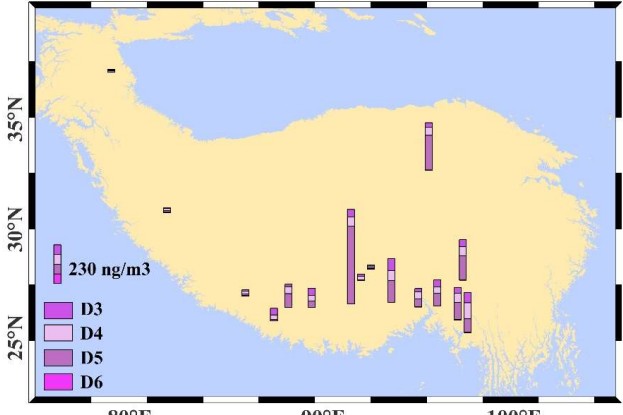

3  **Figure 2 Spatial distribution of cVMS in the atmosphere of the TP**

