# Peer review of "Occurrence and Spatial Distribution of the Neutral Per-fluoroalkyl"

_Atmospheric Chemistry and Physics, 2018_

## Referee Comment (RC1) · Anonymous Referee #1 · 14 Mar 2018

Excellent distribution to the background data of new emerging POPs in Tibet were made. The authors show the occurence and spatial distribution of 14 neutral PFASs and cVMS atmospheric samples from 16 sampling sites in Tibet. Local contribution of cVMS at the capital city of Tipet, the potential long-range atmospheric transport of FTOHs, and elevating concentration of shorter-chain volatile PFAS precursors were highlighted. The concentration of target chemicals in background sites in Tibet were 1~3 orders of magnitude higher than those reported for legacy POPs. This study should be accepted with several following minor revisions.

[Figure]

**[ACPD](ACPD)**

Interactive
comment

1. The authors might want to list the limitations of their study in a paragraph in the Results and Discussion. For example, if other researches asked you for advice on doing a study like this, and they had unlimited resources, what would you tell them to do differently? 2. P3, L61 and 62: need a citation for the restriction of VBS by European Chemical Agency. 3. A few statiscal analysis were performed. It would be better to let readers know how did you design and perform your statiscal analysis? 4. P14, L375: how "poor-relationship" was that betwwen short- and long- chain PFASs? The author should use alternative scientific explainations for this. 5. P7, sample analysis: it would be better if more detail instrumental analysis information was provided in the main manuscrip or supplementary information.

---

## Referee Comment (RC2) · Anonymous Referee #2 · 26 Apr 2018

The manuscript describes multiple pollutant in the atmosphere of the Tibetan Plateau region. The study is also important considering limited literature available on the fate and source identification of PFASs and cyclic volatile Methylsiloxanes from Asia. Certainly, this study will be a valuable addition to present literature on distribution, fate and transport of emerging persistent organic pollutants. Manuscript has presented all facts and explanations in a clear fashion which makes it easy to interpret and understand. This is an important subject and a rising topic and is well suitable for publication on "Atmospheric Chemistry and Physics". I would recommend it minor revision. There

are some suggestions that can help authors improve their manuscript 1. Abstract, line 25: silxoanes should be siloxanes. 2. Lines 71-73: FTOHs and FTOs are not congeners, but compound classes. 4.2 FTOH is a substance which belongs to one of these classes. 3. Line 106-107: there are no estimations of risks in this manuscript. The text in the concluding section suggests that the risks of the "emerging pollutants" are higher without doing any calculations. Do you think the comparison to legacy POPs is meanful? 4. Materials and methods, lines 130-131. The sampling design seems arbitrary. Could you explain a bit about why you choose these sampling sites? 5. Line 190 – Do you refer to recovery of the internal standard? Please clarify. 6. Line 197-198: How could the conversion have happened during sampling? 7. Line 253-254: Can this be caused by phasing-out time? Products containing FOSEs and FOSAs were mostly produced by 3M and mostly phased out in 2002. Products releasing 8:2 FTOH were more recently phased out and the US EPA Stewardship Program only concluded in 2015. 8. Line 356-371: There is no discussion of correlations of 6:2 FTOH with other FTOHs. 9. Line 353-354: This is not clear to me 10. Line 372-380: Are there any correlations between releases of substance and population/wealth where there are a large number of consumer products?
* * *

---

## Author Comment (AC1) · 10 May 2018

Dear respected professor: We appreciate the reviewers' comments, which surely improve our manuscript. According to reviewer's comments, we revised this manuscript carefully. All responses and answers are listed below. All revisions were marked as the highlighted text in the manuscript.

Q1: The authors might want to list the limitations of their study in a paragraph in the Results and Discussion. For example, if other researches asked you for advice on

doing a study like this, and they had unlimited resources, what would you tell them to do differently? A: The limitation of the current study includes, but not limited to, 3 parts: 1) the total numbers of sampling sites (16) were not enough since the Tibetan Plateau is about 250 km2; if resources are unlimited, 40 sampling sites are minimum; 2) sampling rates of SIP were estimated using absorption equation, while they should be calibrated using active air sampler that electricity is needed and air volume is accurate; 3) given south Asia is an important source region and the Yarlung Tsangpo Grand Canyon is a channel for receiving pollutants from southern Asia, deploying a sets of sampler along this Canyon can provide direct evidence regarding the monsoon transport of emerging pollutants, such as neutral per-fluoroalkyl substances, and cyclic volatile methylsiloxanes.

Q2: P3,L61 and 62: need a citation for there striction of VBS by European Chemical Agency. A: A reference was included in line 62 and line 552-555. Please see the following reference: European Chemical Agency (ECHA). Identification of PBT and vPvB substance. Results of evaluation of PBT/vPvB properties for decamethylcyclopentasiloxane; 2012 [Available online at http://echa.europa.eu/documents/10162/13628/decamethyl_pbtsheet_en.pdf].

Q3: A few statistical analysis were performed. It would be better to let readers know how did you design and perform your statistical analysis? A: There are two statistical analysis performed in this study, ANOVA and multiple linear regression, respectivly. Analysis of Variance (ANOVA) consists of calculations that provide information about levels of variability within a regression model and form a basis for tests of significance. Multiple linear regression attempts to model the relationship between two or more explanatory variables and a response variable by fitting a linear equation to observed data. All these two statistical methods are common used methods. In the revision, some brief introductions about the methods are included. "One-way analysis of variance (ANOVA) was performed to determine the statistical differences in the values of individual chemicals among different sampling sites. If the p- value is lower than

0.05, we conclude the significant differences occur." Please see line 308-311. For the multiple linear regressions, the R2 values can be used to explain the variation of the dependents. Please see line 432-433.

Q4: P14, L375: how "poor-relationship" was that between short- and long- chain PFASs? A: Correlation coefficients (r=0.283) between concentrations of NMeFBSE and those of NMeFOSE, obtained by Li et al., (2011) is provided in line 384.

Reference: Li, J., Vento, S. D., Schuster, J., Zhang, G., Chakraborty, P., Kobara, Y., and Jones, K. C.: Perfluorinated Compounds in the Asian Atmosphere, Environ. Sci. Technol., 45, 7241-7246, 2011.

Q5: The author should use alternative scientific explainations for this. A: Sorry about this comment. Since you did not mention the line number, I don't know which place need explaination.

Q6: P7, sample analysis: it would be better if more detail instrumental analysis information was provided in the main manuscrip or supplementary information. A: Detailed analysis regarding GC program and MS detection ions are provided in Text S2.

Thermo DSQ GC–MS was used for analysis of neutral PFASs. The GC temperature program is as follow follows: 50 ℃(held 1 min), ramped at 3 ℃/ min to 70 ℃, ramped at 10 ℃ /min to 130 ℃, then ramped at 20 ℃/ min to 225 ℃ (held 11.4 min), and finally cooled at 80℃/min to50℃ (total run time 32min). A constant injection temperature of 200 ℃ was used, with a 2 mL splitless injection. Helium was employed as a carrier gas at a constant flow of 1 mL/min. The GC–MS transfer line temperature was set at 250 1C. Mass spectral analysis was performed in PCI-selected ion monitoring (PCI–SIM) mode, using methane as the reagent gas for quantification of target analytes (a seven point calibration curve (20, 50, 100, 200, 500, 1000 and 2000 pg injected) was used.

Quantification of cVMS was performed on a Trace GC Ultra (Thermo Electron Corp.)
coupled to a MD800 MS detector (Fisons Instruments SpA) using electron ionization (EI). The GC temperature program incorporated an initial temperature of 40 °C with a hold time of 3 min, increased by 25 °C min−1 to 190 °C, followed by a second temperature ramp of 40 °C min−1 to 240 °C and held for 4 min. The MS was operated in selected ion monitoring (SIM) mode. The following ions were monitored in SIM mode: recording ions m/z 281/282 for D4, 355/267 for D5, 341/429 for D6, 284/285 for 13C4–D4, 360/270 for 13C5–D5, and 345/435 for 13C6-D6, respectively.

Please also see the following references: 1ïijĽ Jonathan L. Barber, et al., Analysis of per- and polyfluorinated alkyl substances in air samples from Northwest Europe. J. Environ. Monit., 2007, 9, 530–541 2ïijĽ Nicholas .A. Warner, et al., Positive vs. false detection: A comparison of analytical methods and performance for analysis of cyclic volatile methylsiloxanes (cVMS) in environmental samples from remote regions. Chemosphere 2013, 93, 749-756 3ïijĽ De-goWang, et al., Determination of cyclic volatile methylsiloxanes in water, sediment, soil, biota, and biosolid using large-volume injection–gas chromatography–mass spectrometry. Chemosphere 2013, 93, 741-748 4ïijĽ Amelie Kierkegaard et al., Determination of linear and cyclic volatile methylsiloxanes in air at a regional background site in Sweden. Atmospheric Environment, 2013, 80, 322-329

Please also note the supplement to this comment:
https://www.atmos-chem-phys-discuss.net/acp-2018-151/acp-2018-151-AC1-supplement.pdf

---

## Author Comment (AC2) · 10 May 2018

Dear Respected Professor: We appreciate the reviewers' comments, which surely improve our manuscript. According to reviewer's comments, we revised this manuscript carefully. All responses and answers are listed below. All revisions were marked as the highlighted text in the manuscript.

Q1. Abstract, line 25: silxoanes should be siloxanes. A: This had been changed. Please see line 25 in revision.

Q2. Lines 71-73: FTOHs and FTOs are not congeners, but compound classes. 4.2 FTOH is a substance which belongs to one of these classes. A: This had been changed. Please see line 72 in revision.

Q3. Line 106-107: there are no estimations of risks in this manuscript. The text in the concluding section suggests that the risks of the "emerging pollutants" are higher without doing any calculations. Do you think the comparison to legacy POPs is meanful? A: We changed this sentence to "Combining the results of this study with the published data regarding legacy POPs in the TP, and emerging POPs in other Asian regions, will provide useful insights to understand the exposure risks of legacy and emerging POPs in the Tibetan environment, and gain a comprehensive understanding of the distribution pattern of emerging POPs in Asia." Please see line 104-108.

In conclusion part, we did not compare exposure risk of emerging and legacy POPs and we just highlighted that concentration of neutral PFASs in the air of the TP are in the hundreds of pg/m3, and levels of cVMS are in the ng/m3 range, which are 2–3 times and 1–2 orders of magnitude, respectively, higher than those for legacy chemicals (such as DDT and HCHs, with maximum concentrations in the tens of pg/m3). Due to the high concentration in air, the continueous emission by local habitants and the poor regulatory in neighbor counties, the risk and harm effect of emerging chemicals should be considered in future. From this perspective, the comparison is meanful.

Q4. Materials and methods, lines 130-131. The sampling design seems arbitrary. Could you explain a bit about why you choose these sampling sites? A: The sampling design is not arbitrary. The sampling sites had been used for monitoring the legacy POPs for around 10 years, and all these sites covers a good spatial coverage of the TP, including 5 sites in the monsoon region; 3 sites in north of 35N, as the westerly domain; and 8 sites located in the transition domain, which is under the control of a shifting climate between Indian monsoon and westerly (Wang et al., 2016).

Reference: Wang et al., Spatial distribution of the persistent organic pollutants across

the Tibetan Plateau and its linkage with the climate systems: a 5-year air monitoring study Atmos. Chem. Phys., 2016, 16, 6901–6911

Q5. Line 190–Do you refer to recovery of the internal standard? Please clarify. A: Yes, here we refer to the recovery of the internal standard. We had clarified this in revision. Please see line 193.

Q6. Line197-198: How could the conversion have happened during sampling? A: We deleted this sentence. Sometimes the recovery above 100% may be caused by erros from extraction and measurements.

Q7. Line 253-254: Can this be caused by phasing-out time? Products containing FOSEs and FOSAs were mostly produced by 3M and mostly phased out in 2002. Products releasing 8:2 FTOH were more recently phased out and the US EPA Stewardship Program only concluded in 2015. A: Thank you for this inspiration. We included this explaination in revision. Please see line 256-259.

Q8. Line 356-371: There is no discussion of correlations of 6:2 FTOH with other FTOHs. A: Some discussion regarding correlations of 6:2 FTOH with other FTOHs were included in revision, please see line 272-277.

Q9. Line 353-354: This is not clear to me A: We reorganized this sentence. "Given that the transition zone is located in the hinterland (central part) of Tibet, where both monsoon and westerly winds become week, and the fresh impact of source regions of either India or Europe/central Asia is limited, thus, the aged/old PFASs in the air of central TP is expected and reasonable." Please see line 360.

Q10. Line 372-380: Are there any correlations between releases of substance and population/wealth where there are a large number of consumer products? A: Sorry, data regarding the amount of consumer products in Tibet and India cannot be gotten from literatures or other documents, we are not able to conduct correlations analysis.

Please also note the supplement to this comment:
https://www.atmos-chem-phys-discuss.net/acp-2018-151/acp-2018-151-AC2-supplement.pdf